# Ultrasound-Guided Multi-Branch Rectus Femoris Nerve Block for Spasticity Assessment

**DOI:** 10.3390/toxins17090437

**Published:** 2025-09-01

**Authors:** Stefano Carda, Elisa Grana, Thierry Deltombe, Rajiv Reebye

**Affiliations:** 1Service Universitaire de Neurorehabilitation, Lausanne University Hospital (CHUV), Rue du Bugnon 46, Av. Pierre-Decker 5, 1011 Lausanne, Switzerland; elisa.grana@chuv.ch; 2Department of Physical Medicine and Rehabilitation, CHU UCL Namur Site Godinne, Avenue Dr G. Therasse 1, 5530 Yvoir, Belgium; thierry.deltombe@chuuclnamur.uclouvain.be; 3Division of Physical Medicine and Rehabilitation, Faculty of Medicine, University of British Columbia, 2775 Laurel Street, Vancouver, BC V5Z 1M9, Canada; rajiv.reebye@vch.ca

**Keywords:** diagnostic nerve block, rectus femoris, spasticity, stiff knee gait, ultrasound guidance, femoral nerve

## Abstract

**Background**: Stiff-knee gait commonly involves rectus femoris spasticity in patients with central nervous system lesions. Diagnostic nerve blocks aid in predicting treatment outcomes; however, current techniques may overlook multiple nerve branches that innervate the rectus femoris muscle, potentially resulting in an incomplete assessment of treatment outcomes. **Methods**: We present an ultrasound-guided approach that we currently use in our practice, using anatomical landmarks, including the femoral artery, the sartorius muscle, and the rectus femoris’ characteristic “J-shaped” internal tendon. The technique employs an “elevator” scanning method to identify all motor nerve branches (typically 2–3) entering the proximal third of the rectus femoris muscle. Each branch is blocked using an in-plane needle approach with 1–2 mL of 2% lidocaine. **Results**: The technique enables the visualization of hyperechoic nerve branches entering the rectus femoris muscle from medial to lateral, sometimes accompanied by small vascular branches that are identifiable with a Doppler ultrasound. Optimal ultrasound settings include probes >8 MHz, appropriate focus positioning, and dynamic range < 60 dB. The multi-branch approach produces rapid-onset motor weakness (5–10 min). **Conclusions**: This comprehensive multi-branch rectus femoris nerve block technique may enhance diagnostic accuracy for spasticity assessment, potentially leading to more informed treatment selection for stiff-knee gait.

## 1. Introduction

Stiff knee gait is one of the most prevalent pathological gait patterns following central nervous system lesions [1]. It is fundamentally characterized by reduced knee flexion during the swing phase of the gait [2]. This condition can significantly impact locomotion for patients with stroke, traumatic brain injury, cerebral palsy, and other neurological conditions affecting motor control.

The rectus femoris (RF) muscle has emerged as a critical focus in understanding and managing stiff-knee gait due to its unique dual function as both a hip flexor and a knee extensor. This anatomical characteristic causes inappropriate activation during the swing phase detrimental to physiological gait mechanics [3,4]. When the RF muscle exhibits spasticity or abnormal timing during the swing phase, it can prevent adequate knee flexion, leading to compensatory mechanisms such as hip circumduction and/or excessive plantar flexion on the contralateral side, which elevates the center of body mass to clear the foot during swing.

Contemporary evidence suggests that RF spastic co-contraction significantly contributes to the stiff-knee gait pathophysiology, with electromyographic studies revealing abnormal timing patterns in up to 96% of stroke patients exhibiting this gait deviation [5]. However, the clinical assessment of RF’s contribution to stiff-knee gait remains challenging. Traditional clinical bed tests, such as the Duncan-Ely test, have limited reliability in predicting RF overactivity during gait [6], indicating that more complex diagnostic approaches are necessary to accurately assess the relative contribution of RF spasticity to stiff knee. Diagnostic motor nerve blocks using local anesthetics have emerged as valuable tools for temporarily interrupting nerve conduction in order to reduce muscle spasticity and co-contraction, thereby allowing clinicians to assess the specific contribution of individual muscles to pathological gait patterns [7,8]. Nerve blocks can provide relevant information for treatment planning, particularly when considering interventions such as chemodenervation, surgical lengthening (rectus transfer), or selective neurotomy.

The temporary nature of the blockade allows for objective gait analysis before and after the procedure, providing clear evidence of the muscle’s contribution to the observed gait deviation.

Previously described approaches to RF nerve block have relied primarily on anatomical landmarks and on the use of a neuromuscular stimulator [9,10]. While these methods have demonstrated clinical utility, they possess inherent limitations related to anatomical variability and the potential for incomplete blockade. Recent advances in ultrasound use have facilitated the development of ultrasound-guided techniques as effective alternatives to landmark-based approaches [9,11]. Ultrasound guidance offers several advantages, including real-time visualization of anatomical structures, direct observation of needle placement, and confirmation of local anesthetic spread around target nerves.

However, a critical limitation of existing RF nerve block techniques, whether landmark-based or ultrasound-guided, is their focus on blocking a single nerve branch, typically the main proximal nerve to the RF muscle [9,11,12]. This approach fails to account for the well-documented anatomical variability in femoral nerve branching patterns to the RF muscle. Anatomical studies have consistently demonstrated that the femoral nerve provides 1–4 branches to the RF muscle, with 2–3 branches being the most common configuration [13]. This anatomical variability suggests that single-branch blockade techniques may result in incomplete RF paralysis or diffusion to vastus motor nerve branches [11] (Authors’ opinion), potentially leading to an inaccurate assessment of the muscle’s contribution to stiff-knee gait.

The incomplete blockade from single-branch techniques may have different implications. If residual RF function persists due to an insufficient block, the diagnostic value of the procedure is reduced. Conversely, a comprehensive blockade that addresses all nerve branches would provide more accurate information about the actual contribution of RF spasticity to the patient’s gait dysfunction.

The present study addresses this limitation by describing a novel ultrasound-guided technique for comprehensive RF motor nerve block that systematically targets all existing nerve branches. This paper is a procedural description to show the technique and has not been formally tested against current techniques.

This approach can improve diagnostic nerve block methodology, potentially enhancing the accuracy of spasticity assessment and treatment planning for patients with stiff knee gait.

## 2. Materials and Methods, Procedure and Results

### 2.1. Patient Positioning and Equipment

The patient is supine, with the target extremity positioned in slight hip abduction (10–20 degrees) and 20° external rotation. This positioning optimizes visualization of the femoral neurovascular bundle and surrounding anatomical structures while providing comfortable access for the ultrasound transducer and needle insertion. The examination room should be adequately lit, and the ultrasound monitor positioned to allow clear visualization by the operator without compromising ergonomics.

The ultrasound equipment consists of a high-frequency linear ultrasound transducer with a frequency greater than 8 MHz. Higher-frequency transducers provide superior resolution for superficial structures, which is essential for accurately identifying the relatively small motor nerve branches to the RF muscle. The ultrasound system should have color Doppler capability to identify small vascular branches that may accompany motor nerves.

A 22-gauge insulated needle is connected to a neuromuscular stimulator for the nerve block procedure. The insulated design helps prevent inadvertent stimulation of adjacent structures. Electrical stimulation is recommended to ensure that the structure identified as a nerve branch is not merely a fascial or connective tissue structure. Many patients with spasticity exhibit various degrees of muscle fibrosis, which can sometimes complicate the recognition of the nerve branch with sufficient certainty using ultrasound alone. The stimulation is provided at an intensity of <0.3 mA with a duration of 1 ms. An electromyographic RF H-reflex monitoring can be of added value. The local anesthetic consists of 2% lidocaine, which is suitable for diagnostic purposes. The volume of lidocaine injected is typically between 1 and 2 mL per branch. This allows for the injection of a dose that is safe according to current recommendations for an adult [14].

### 2.2. Ultrasound Technique and Anatomical Identification

We suggest the following setting: (1) a linear probe, with frequency greater than 8 MHz, (2) focus positioned at or 0.5–1 cm deeper than the target depth, and (3) Dynamic Range inferior than 60 dB (this will increase the contrast of the image making the nerve and fascial structures more visible) [15].

The initial step involves identifying the femoral artery, which is the primary anatomical landmark. The femoral artery appears as a round-shaped, non-compressible, pulsating structure on ultrasound examination. Adjacent to the femoral artery, the femoral vein(s) can be identified as compressible vascular structures. The femoral nerve is typically lateral to the femoral artery and appears hyperechoic and fascicular (Figure 1 and Appendix A). The sartorius muscle is identified as it covers the femoral neurovascular bundle, exhibiting a characteristic “reversed canoe” shape in cross-sectional view. This muscle serves as an essential landmark for orientation and helps define the medial boundary of the scanning area. Just lateral to the sartorius muscle lies the RF muscle, which can be reliably identified by its characteristic “J-shaped” internal tendon (Figure 2).

### 2.3. Motor Nerve Branch Identification

Motor nerve branches to the RF muscle appear as hyperechoic, linear structures, measuring 1–2 mm in diameter, with characteristic fascicular patterns. These branches enter the muscle from a medial to lateral direction and are most easily identified using the “elevator” scanning technique. This technique involves systematic scanning from cranial to caudal directions of the target structure (in this case, the rectus femoris muscle), allowing for comprehensive visualization of all nerve branches entering the proximal third of the RF muscle.

The “elevator” technique is performed by placing the ultrasound transducer over the short axis of the RF muscle and systematically moving the probe from cranial to caudal (and vice versa). This approach ensures that all nerve branches are identified, as they may enter the muscle at slightly different levels. The operator should pay particular attention to the medial aspect of the RF muscle, where nerve branches typically penetrate the muscle belly [16].

Small vascular branches often accompany motor nerves and can be visualized using color Doppler ultrasound. When present, these vascular structures create a characteristic “bird’s eye” appearance on ultrasound, with the nerve appearing as the central hyperechoic structure surrounded by the vascular signal. This finding can help confirm the identification of neural structures and distinguish them from other hyperechoic tissues.

The most common anatomical pattern involves 2–3 distinct nerve branches, although variation can occur from 1 to 4 branches [13]. Each identified branch should be carefully mapped, and its entry point into the RF muscle should be noted. The operator should document the number and location of identified branches to ensure comprehensive blockade.

### 2.4. Injection Technique

Each identified nerve branch is approached using an in-plane needle insertion technique from lateral to medial. This approach provides optimal visualization of the needle throughout its trajectory, allowing for precise placement of the needle tip adjacent to each nerve branch. The in-plane technique also minimizes the risk of inadvertent vascular puncture or injury to adjoining structures.

A needle specially designed for conduction anesthesia is advanced under direct ultrasound guidance until the tip is positioned immediately adjacent to the target nerve branch. The needle tip should be placed in the tissue plane surrounding the nerve rather than within it. Electrical stimulation is used to induce a muscle contraction, visible both clinically and with ultrasound, confirming the needle’s appropriate position.

Local anesthetic injection consists of 1–2 mL of 2% lidocaine around each identified nerve branch [17]. The volume may be adjusted based on the size of the nerve branch and the ease of anesthetic spread. During injection, the operator should observe the characteristic hypoechoic spread of local anesthetic around the nerve branch, which confirms appropriate needle placement and adequate anesthetic distribution (Figure 3 and Appendix A).

The injection should be performed slowly to minimize patient discomfort and allow for real-time monitoring of anesthetic spread. If resistance is encountered during injection, the needle position should be reassessed, as this may indicate intraneural placement or contact with fascial planes that could impede the distribution of the anesthetic.

### 2.5. Procedure Documentation and Assessment

The entire procedure should be documented, including the number of nerve branches identified, their anatomical locations, and the volume of local anesthetic used for each branch. Digital images or video clips of the ultrasound findings can be valuable for documentation and future reference.

Following the completion of the nerve block, patients should be monitored for the onset of spasticity reduction/disappearance, which typically occurs within 5–10 min of the injection.

The procedure typically takes 15–20 min, depending on the complexity of the anatomical pattern and the operator’s experience. The technique is relatively easy to learn.

## 3. Discussion

### Addressing Fundamental Limitations of Existing Techniques

This innovative ultrasound-guided multi-branch RF nerve block technique addresses the core limitations of existing single-branch approaches to diagnostic nerve blockade: an incomplete block.

The anatomical basis for this method is well-established in the literature, with multiple studies showing the variable branching patterns of the femoral nerve to the RF muscle [10]. However, earlier nerve block techniques have mostly overlooked this anatomical variability, focusing instead on blocking the most prominent or easily identifiable nerve branch. This approach inherently risks incomplete blockade or sometimes diffusion to the femoral motor nerve branches innervating the vastus intermedius, lateralis, and medialis muscles, which can lead to an inaccurate assessment of the RF’s role in pathological gait patterns.

The clinical effects of incomplete nerve blockade extend beyond diagnostic mistakes. When residual RF function remains because of unblocked nerve branches, clinicians might underestimate the muscle’s role in stiff-knee gait, potentially leading to incorrect treatment choices. Conversely, if a stiff knee gait persists after a complete RF nerve block, it may help identify other causes, like hip flexor muscle weakness or spastic co-contraction of the vastus intermedius. On the other hand, diffusion to nearby motor nerve branches that supply the vastus muscles could overestimate the part of RF spasticity in the stiff knee gait. Additionally, motor block using a multi-branch approach may provide a clearer understanding of the potential benefits of longer-lasting treatments, such as botulinum toxin injections or neurotomy.

The ultrasound-guided method provides several clear benefits over traditional landmark-based techniques for RF nerve blockade. Real-time visualization of anatomical structures significantly reduces the uncertainty associated with landmark-based approaches. Being able to directly see needle placement and the spread of local anesthetic offers immediate feedback on the success of the blockade.

## 4. Conclusions

This ultrasound-guided multi-branch RF nerve block technique may significantly advance the diagnostic nerve block methodology for assessing spasticity. The method addresses the fundamental limitation of existing single-branch approaches by systematically identifying and blocking all motor nerve branches to the RF muscle. It provides enhanced diagnostic accuracy for assessing the RF contribution to stiff-knee gait.

The principles underlying this approach also have broader applications for other muscle groups affected by spasticity, advancing the field of spasticity management beyond the treatment of stiff-knee gait.

## Figures and Tables

**Figure 1 toxins-17-00437-f001:**
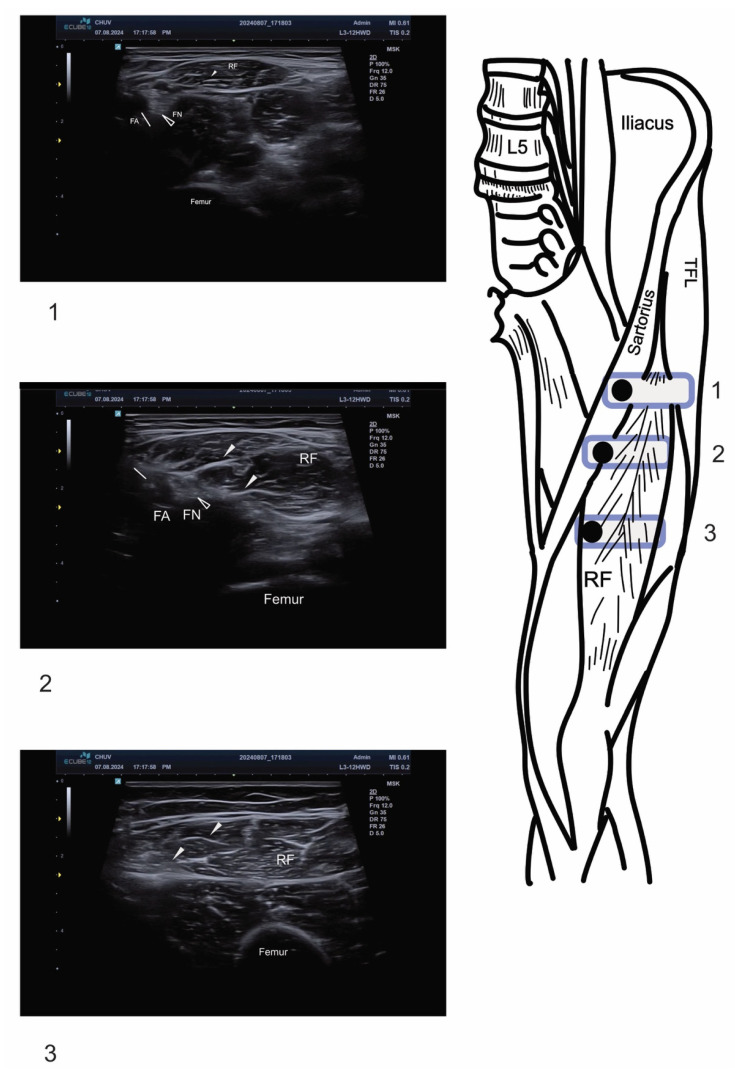
Ultrasound anatomy for multi-branch rectus femoris nerve block. On the right of the figure, the positioning of the linear probe (rounded-corner rectangle) with the corresponding ultrasound image on the left. Black dots within the rectangle represent the upper left corner of the ultrasound image. RF: rectus femoris; FA and white arrow: femoral artery; FN and empty arrowhead: femoral nerve; TFL: tensor fascia lata. Arrowheads indicate intramuscular motor branches of the femoral nerve entering the rectus femoris muscle.

**Figure 2 toxins-17-00437-f002:**
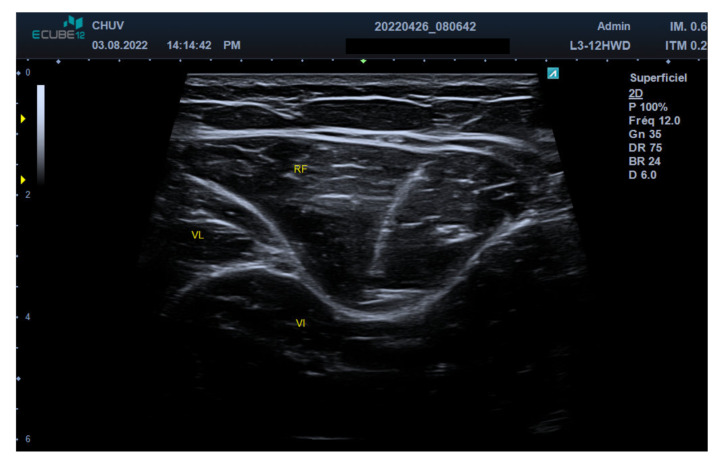
“J”-shaped internal tendon of Rectus Femoris. RF: Rectus Femoris Muscle; VL: Vastus Lateralis Muscle; VI: Vastus Intermedius Muscle.

**Figure 3 toxins-17-00437-f003:**
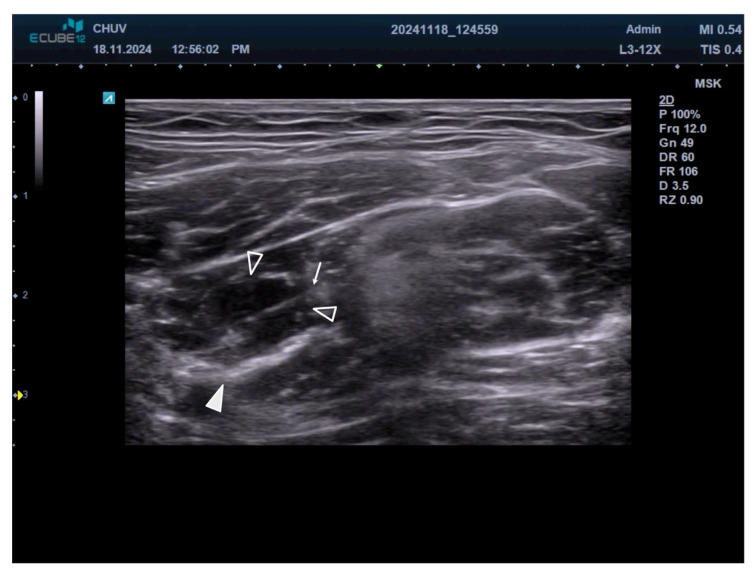
Needle insertion technique and local anesthetic distribution. The arrow shows the needle from lateral to medial. Arrowhead shows the motor branch of the femoral nerve. Empty arrowheads show the diffusion of local anesthetic around femoral nerve motor branches. The hypoechoic spread of anesthetic confirms appropriate needle placement and adequate nerve blockade.

## Data Availability

The original contributions presented in this study are included in the article/Appendix A. Further inquiries can be directed to the corresponding author.

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
