# Peer review of "Ultrasound-Guided Multi-Branch Rectus Femoris Nerve Block for Spasticity Assessment"

_toxins, 2025, doi:10.3390/toxins17090437_

Round 1
Reviewer 1 Report
Comments and Suggestions for Authors
Reviewer Comments:
Significance and Practical Application
The content of this manuscript is of notable significance, particularly as it presents a novel technique. To enhance the impact and clarity of the paper, it is recommended that the authors include information regarding the application of this technique in actual clinical cases, if such data are available.
Title Clarity
The current title could be revised to improve reader comprehension. A more accessible and descriptive title, such as "Ultrasound-Guided Multi-Branch Rectus Femoris Nerve Block for Spasticity Assessment", would more clearly convey the subject matter of the manuscript.
Figure Presentation and Readability
Ultrasound imaging plays a central role in this study. Figure 1 outlines the scanning procedure in detail; however, it is a fundamental requirement that all figures be interpretable without reference to the main text. In this regard, the ultrasound image on the left in Figure 1 lacks sufficient clarity. The authors are advised to improve its visibility and consider adding a clear illustration for guidance.
In Figure 2, the layout should be standardized: the illustration should appear on the left and the ultrasound image on the right. Additionally, explanatory lines or annotations should not be superimposed on the ultrasound image. Instead, the corresponding illustration should guide the interpretation. All figures should be independently understandable, and all abbreviations should be spelled out to ensure clarity. The same principles should be applied to Figure 3.
Clarification of “Elevator Scan”
The term “elevator scan” requires further clarification. The authors should provide a precise description of this scanning method, including how the term relates to the physical movements or technique used during ultrasound imaging.
Discussion Section
While the core message of the paper is clearly understood, the discussion section is somewhat redundant and, at times, difficult to follow. It is recommended that this section be revised for conciseness and clarity to better support the overall argument.
References and Editorial Issues
The references cited appear appropriate. However, there is a sentence in reference 9 that seems unrelated and appears to have been inadvertently inserted:“No commercial party having a direct financial interest in the results of the research supporting this article has or will confer a benefit upon the author(s) or upon any organization with which the author(s) is/are associated.” This sentence should be removed.
The reference 16 is not present in the main text.
Furthermore, the authors are encouraged to thoroughly proofread the revised manuscript to eliminate typographical or careless errors prior to resubmission.
Comments on the Quality of English LanguageThis paper is poorly written, and it isn't easy to understand its content in its current state.
Reviewer 2 Report
Comments and Suggestions for Authors
See attached file

can be improved
Round 2
Reviewer 1 Report
Comments and Suggestions for Authors
The authors responded properly to the peer review comments, which significantly improved the revised manuscript and will enhance reader understanding.